# The Role of Inositols in the Hyperandrogenic Phenotypes of PCOS: A Re-Reading of Larner’s Results

**DOI:** 10.3390/ijms24076296

**Published:** 2023-03-27

**Authors:** Valeria Fedeli, Angela Catizone, Alessandro Querqui, Vittorio Unfer, Mariano Bizzarri

**Affiliations:** 1Department of Experimental Medicine, University La Sapienza, Via A. Scarpa 16, 00160 Rome, Italy; 2Section of Histology and Embryology, Department of Anatomy, Histology, Forensic Medicine and Orthopedics, Sapienza University of Rome, Via A. Scarpa 16, 00160 Rome, Italy; 3Systems Biology Group Lab, University La Sapienza, 00185 Rome, Italy; 4The Experts Group on Inositol in Basic and Clinical Research (EGOI), 00161 Rome, Italy; 5AGUNCO Obstetrics & Gynecology Center, UniCamillus-Saint Camillus International University of Health and Medical Sciences, 00131 Rome, Italy

**Keywords:** polycystic ovary syndrome, D-Chiro-inositol, myo-inositol, insulin resistance, epimerase, hyperandrogenism

## Abstract

Polycystic ovarian syndrome (PCOS) is the most common endocrinological disorder in women, in which, besides chronic anovulation/oligomenorrhea and ovarian cysts, hyperandrogenism plays a critical role in a large fraction of subjects. Inositol isomers—myo-Inositol and D-Chiro-Inositol—have recently been pharmacologically effective in managing many PCOS symptoms while rescuing ovarian fertility. However, some disappointing clinical results prompted the reconsideration of their specific biological functions. Surprisingly, D-Chiro-Ins stimulates androgen synthesis and decreases the ovarian estrogen pathway; on the contrary, myo-Ins activates FSH response and aromatase activity, finally mitigating ovarian hyperandrogenism. However, when the two isomers are given in association—according to the physiological ratio of 40:1—patients could benefit from myo-Ins enhanced FSH and estrogen responsiveness, while taking advantage of the insulin-sensitizing effects displayed mostly by D-Chiro-Ins. We need not postulate insulin resistance to explain PCOS pathogenesis, given that insulin hypersensitivity is likely a shared feature of PCOS ovaries. Indeed, even in the presence of physiological insulin stimulation, the PCOS ovary synthesizes D-Chiro-Ins four times more than that measured in control theca cells. The increased D-Chiro-Ins within the ovary is detrimental in preserving steroidogenic control, and this failure can easily explain why treatment strategies based upon high D-Chiro-Ins have been recognized as poorly effective. Within this perspective, two factors emerge as major determinants in PCOS: hyperandrogenism and reduced aromatase expression. Therefore, PCOS could no longer be considered a disease only due to increased androgen synthesis without considering the contemporary downregulation of aromatase and FSH receptors. Furthermore, these findings suggest that inositols can be specifically effective only for those PCOS phenotypes featured by hyperandrogenism.

## 1. Hyperandrogenism: A Pivotal Role in the Pathogenesis of the Polycystic Ovary Syndrome

Polycystic ovarian syndrome (PCOS) is the most common cause of anovulation [1], initially identified as an association of chronic anovulation/oligomenorrhea, with clinical or biochemical hyperandrogenism [2]. The exact cause of PCOS is unknown, albeit several metabolic, endocrine, and genetic factors have been credited to play a significant role. A cornerstone was set in 2003, when the Rotterdam consensus revised the diagnostic criteria, indicating two of three of the following criteria as mandatory for PCOS diagnosis: chronic anovulation or oligomenorrhea, clinical or biochemical hyperandrogenism, and polycystic ovarian morphology [3]. The Rotterdam statement has prompted an intense debate, finally leading to an updated version of the previous criteria. Currently, we distinguish four different phenotypes: phenotype A with the coexistence of clinical hyperandrogenism/hyperandrogenemia (HA); oligomenorrhea/anovulation (OA), and polycystic ovary morphology (P); phenotype B with HA and OA without P; phenotype C with HA and P and regular ovulatory cycles; and phenotype D, where OA coexists with P in the absence of HA [4]. Noticeably, hyperandrogenism and anovulation are tightly linked to insulin resistance [5]. Deregulation of insulin signaling not only explains several metabolic aspects of PCOS—independently of the associated obesity—but contributes to modulating ovarian steroidogenesis towards a stable “androgenic” phenotype [6]. 

Women suffering from PCOS—excluding the phenotype D—share several clinical features of hyperandrogenism, including hirsutism, acne, and alopecia [7]. Excessive androgen production, either by adrenals or by ovaries—eventually by a combination of both—play a relevant role in the pathogenesis of PCOS. From a systemic point of view, abnormalities in the hypothalamic–hypophysis–adrenal axis (HPA)—triggered by several physical/endocrine cues—can alter the properly balanced pulse frequency of gonadotropin-releasing hormone, ultimately leading to an increased LH/FSH ratio [8]. The imbalance in LH/FSH release contributes to enacting the proliferation of ovarian theca cells, leading to increased androgen synthesis, given that LH stimulates ovarian androgen production, whereas a relative deficit in FSH impairs follicular development. Furthermore, increased insulin levels and insulin resistance amplify androgen synthesis from both adrenals and theca cells of the ovary [9]. However, a compelling body of evidence suggests that ovarian hyperandrogenism should be primarily ascribed to an intrinsic steroidogenic defect of theca cells in PCOS [10], which in turn could be amplified by several systemic factors, including deregulation of the crosstalk between HPA and the ovaries [9,11], metabolic and microbiota factors [12,13], and insulin resistance. 

## 2. A Light in the Dark: The myo-Ins/D-Chiro-Ins Connection

A thought-provoking paper published almost twenty years ago reported very intriguing results [14]. In PCOS patients with classical PCOS features, normal insulin levels, and metabolic insulin sensitivity, the pharmacological reduction in insulin secretion induced a significant decrease in blood testosterone levels. Of note, in the same cluster of patients, total suppression of LH levels following the administration of long-acting GnRH agonist leuprolide acetate was not as effective as the inhibition of insulin secretion in blocking androgen production. This study clearly indicated that, even in the absence of overt insulin resistance, the ovaries of PCOS women display an intrinsic sensitivity to normal endocrine signaling (insulin and LH), which is ultimately responsible for increased androgen release.

How could we explain this abnormal sensitivity? A significant contribution in explaining this puzzle has been provided by Larner’s seminal paper demonstrating that ovarian abnormal response to insulin alters the specific myo-Inositol/D-Chiro-Inositol (myo-Ins/D-Chiro-Ins) ratio within the ovary [15]. Previous studies have shown that diabetes and insulin resistance is associated with the reduced transformation of myo-Ins into D-Chiro-Ins in high-glucose-consuming tissues. This abnormal inositol pattern is primarily ascribed to the impaired conversion mediated by tissue-specific epimerase, activated under insulin stimulation. Impairment of insulin signaling, which characterizes insulin resistance, will then inhibit myo-Ins conversion into its epimer [16]. D-Chiro-Ins is usually incorporated into glycosylphosphatidylinositol-anchored proteins (GPI), from which it is released as inositol-phosphoglycan (IPG-P), an intermediary metabolite displaying a critical role in transducing insulin effects [17]. A ‘functional’ increase in IPG-P values has been observed following an oral glucose challenge in healthy individuals, but not in subjects with diabetes mellitus [18]. Namely, needle biopsies, as well as autopsy studies performed on muscle mass from subjects suffering from diabetes, confirmed a significantly reduced content of D-Chiro-Ins with respect to control samples [19,20]. In turn, it has been documented that reduced availability of D-Chiro-Ins is followed by several defects in the proper utilization of glucose [21], and the myo-Ins/D-Chiro-Ins ratio has been suggested as a measure of insulin resistance [22].

Nevertheless, it is well established that ovarian cells from PCOS subjects retain insulin sensitivity, even in those patients showing insulin resistance, as both insulin and hyperinsulinemia still stimulate ovarian androgen production in PCOS [23]. Therefore, it would be expected that insulin could increase myo-Ins conversion into D-Chiro-Ins within the ovary, even in insulin-resistant women. Indeed, this is the case, as shown by the previously quoted study from Larner’s team. In theca cells from both normal and PCOS women, myo-Ins, D-Chiro-Ins levels, and epimerase activity have been investigated, showing that epimerase activity was three times higher than normal in PCOS subjects, while the myo-Ins/D-Chiro-Ins ratio was four times lower in PCOS subjects when compared to normal theca cells. It is worth noting that, in this study, theca cells were stimulated by using the same quantity of insulin [15]. Therefore, those results demonstrated that theca cells from PCOS subjects not only retain their insulin sensitivity, but they transform myo-Ins into D-Chiro-Ins with higher efficiency than normal theca cells, even under the same insulin stimulation. Consequently, PCOS ovarian cells show a paradoxical insulin “hypersensitivity” compared to peripheral tissues. Indeed, analyses of inositol content in follicular fluid obtained from PCOS patients vindicated this hypothesis, by establishing that the myo-Ins/D-Chiro-Ins ratio dramatically decreases in PCOS subjects with respect to controls. In fact, while the myo-Ins/D-Chiro-Ins ratio is nearly 100:1 in normal subjects, in the follicular fluid of PCOS women, that value is barely 0.2:1 [24].

Moreover, the increase in D-Chiro-Ins availability within the ovary following insulin stimulation would provide increased chiro-inositol to be incorporated into precursor GPI-phospholipid and or precursor GPI-protein. Insulin activates phospholipase anchored to the external layer of cell membranes [25], thus further increasing the release of IPGs, the intermediate messenger displaying several ‘insulin-mimetic’ activities. Therefore, we posit that by raising ovarian D-Chiro-Ins, the insulin signaling would be amplified [26,27] 

## 3. Consequences of Increased D-Chiro-Ins Synthesis in the Ovary

These results have outstanding consequences, given that IPGs have been shown to be required for androgen synthesis downstream of insulin stimulation [28]. Noticeably, increased IPG availability explains why insulin’s action on PCOS theca cells is massively greater than on controls in increasing testosterone synthesis [29]. However, consequences of increased D-Chiro-Ins availability are not restricted to androgen synthesis in theca cells and involve granulosa cells as well. 

A study performed on granulosa cells obtained from PCOS women has shown that D-Chiro-Ins reduces the mRNA expression of both aromatase and cytochrome P450 side-chain cleavage genes in a dose–response fashion [30]. Preliminary data from our laboratory indeed confirm that D-Chiro-Ins enhances the activity of key androgenic enzymes (such as 3-βHSD) while specifically inhibiting aromatase (Cyp19A1) synthesis and FSH receptor (FSHr) release in ovary cells obtained from PCOS animals. These preliminary findings indicate that D-Chiro-Ins acts as a double-edged sword, by increasing androgen production from theca cells while inhibiting the estrogenic response of granulosa cells.

These recent data can help explain some puzzling clinical results provided by treating PCOS patients with high doses of D-Chiro-Ins. In a study authored by Nestler’s group, a positive correlation between changes in IPG-P release and in insulin sensitivity in PCOS patients treated with very high levels of D-Chiro-Ins (1500 mg twice daily for 6 weeks) was observed [31]. Unfortunately, that investigation was incapable of confirming the beneficial effects on ovary function previously obtained by the same team with low doses of D-Chiro-Ins in PCOS patients [32]. Moreover, high D-chiro-Ins levels negatively influence the quality of oocytes and blastocysts [33]. Conversely, treatment with D-Chiro-Ins (1 g/day) on male volunteers led to a dramatic reduction in both estrone and 17β-estradiol levels (−85.0% and −14.4%, respectively), while testosterone and dehydroepiandrosterone became highly increased (+23.4% and +13.8%, respectively) [34]. These results suggest that D-Chiro-Ins unexpectedly displays anti-estrogenic effects, likely through inhibition of aromatase activity. This property is indeed currently under study for the treatment of male and female clinical conditions that would benefit from an androgen increase and/or estrogen decrease [35]. Based on those results, D-Chiro-Ins should be deemed detrimental in PCOS subjects, given that aromatase functional deficiency plays a critical role in the pathogenesis of the syndrome [36,37]. In fact, since the nineties, some reports anticipated that the follicular fluid of PCOS women “contains one or more endogenous inhibitors of aromatase activity” [38]. It is likely that D-Chiro-Ins could perform precisely such a role. Indeed, this hypothesis has been investigated on wild-type female mice treated with increasing doses of D-Chiro-Ins for almost five ovulatory cycles [39]. Changes in ovarian histology and tissue expression of both aromatase and testosterone were analyzed. At any rate, some caution is needed in extrapolating data obtained from models in which PCOS is experimentally triggered by using either androgen administration or continuous light exposure; results provided by the aforementioned study demonstrated that high doses of D-Chiro-Ins supplementation profoundly altered ovarian histology, increased serum testosterone levels, and reduced aromatase levels. Overall, these data indicate that, contrary to preliminary expectations [32], D-Chiro-Ins exerts a detrimental effect on ovary steroidogenesis.

Nonetheless, these assumptions have been challenged for a while by controversial clinical results obtained in PCOS patients treated with inositol formulations, in which both myo-Ins and D-Chiro-Ins have been associated according to different ratios [40]. 

Again, this is a paradox that prompted us to reconsider the basic physiology of inositol to assess if inositol isomers exert either complementary or opposite actions upon ovarian steroidogenesis.

## 4. Inositols in Mammals 

In the middle of the 19th century, Johann Joseph Scherer extracted a polyol-a hexa-hydroxy-cyclohexane compound—hence named “inositol”—from muscle cells [41]. The chemical structure was ascertained later, and the configuration of all of its related isomers only recently has been clearly described [42]. Noticeably, the hexahydroxycyclohexane backbone displays a bewildering plasticity in allowing the emergence of nine different isomers: cis-, epi-, allo-, myo-, neo-, scyllo-, L-chiro-, D-Chiro-, and muco-inositol. Myo-Inositol is by far the most abundant form in nature, while two other ‘minor’ isomers—scyllo- and D-Chiro-Ins—have been shown to exert an appreciable biological function [43]. It is now well established that myo-Ins is an essential component in many physiological processes and biochemical reactions. However, it is still difficult to identify a clear metabolomic profile [44], as myo-Ins in the organism derives from different sources—exogenous and endogenous—and it is further metabolized according to an intricate network of biochemical transformations. 

In humans, myo-Ins (~1 g/day) primarily comes from dietary intake, albeit a relevant fraction is synthesized endogenously, in a variable range (from 1 to 4 g/day), depending on several factors, yet requiring a full assessment [45]. Within the mammalian cell, myo-Ins is synthesized from glucose-6-phosphate (G6P), which is isomerized to inositol-3-phosphate (Ins-(3)-P) by D-3-myo-inositol-phosphate synthase (inositol synthase, MIPS1) [46], encoded by the inositol-3-phosphate synthase 1 (ISYNA1) gene [47]. Then, inositol monophosphatase-1 (IMPA1 or IMPase) dephosphorylates Ins-(3)-P to release free myo-Ins [48]. The ISYNA1 is an essential gene for proper replication and differentiation, as knockout cells cultured in inositol-free media cannot proliferate, whereas ISYNA1^−/−^ animals showed a general modification in the global gene expression profile and a significant increase in embryonic lethality [49]. It is worth noting that, while in yeasts, for the INO1 gene—the homologs of ISYNA1—the environmental availability of inositol regulates inositol-3-phosphate synthesis from G6P through the modulation of the Ino2/Ino4 complex that “senses” inositol concentrations [50], in mammals, ISYNA1 transcription seems to rely principally upon internal cellular signals [51]. Glucose availability plays a pivotal role, as glucose shortage activates AMPK that, in association with phosphatidic acid (PA)—released by cell membranes—downregulates ISYNA1 expression, and hence MIPS1 activity [52]. This effect is dependent on the nuclear translocation of IP6K1—an enzyme responsible for the conversion of inositol hexakisphosphate (InsP6) into diphosphoinositol pentakisphosphate (InsP7/PP-InsP5)—that senses changes in the ATP/ADP ratio [53]. Reduced glucose availability will then inhibit endogenous myo-Ins synthesis, and consequently, it is tempting to speculate that myo-Ins metabolism would be likely “disturbed” in several conditions, including diabetes, insulin resistance, and cancer, in which cellular “avidity” for glucose is raised.

The cell’s machinery uses myo-Ins to build up several derived inositol-phosphates (InsPs) as well as complex macromolecules, including phosphatidyl-inositol (PI), its seven phosphoinositides (phosphatidyl-inositol phosphate, PIP), and glycosylphosphatidylinositol-anchored proteins (GPI), localized mostly at the surface of cell membranes [54]. Myo-Ins is incorporated into eukaryotic cell membranes as phosphatidyl-myo-inositol. Hence, the inositol ring can be phosphorylated by a variety of kinases on the three, four, and five hydroxyl groups in seven different combinations, leading to three main phosphoinositides: PIP1, PIP2, and PIP3. A pivotal hub within that network is constituted by phospathidyl-Inositol-4,5-phosphate (PIP2), which can be hydrolyzed by phospholipase C (PLC), then splitting PIP2 into 1,2-diacylglycerol (DAG) and InsP3, a key, second messenger in the transduction of several endocrine signals [55]. Inositol 1,4,5-trisphosphate (Insp3) binds to specific InsP3-receptors, inducing calcium release from the endoplasmic reticulum. As for other second messengers, InsP3 has a short half-life, and it is rapidly metabolized through one of the two following pathways [56]: removal of the 5-phosphate from the inositol ring by Ins-polyphosphate 5-phosphatases, resulting in the release of InsP2, and successively of InsP1, before being finally dephosphorylated to reconstitute free myo-Ins. On the contrary, very different biochemical pathways lead to successive phosphorylation of InsP3 to produce InsP4, under the action of Ins(1,4,5)P3 3-kinase [57]. Successively, 1,3,4,5,6-pentakisphosphate 2-kinase [58] catalyzes the synthesis of InsP5 and InsP6, albeit other enzymes–IPK1 and IPK2–can promote InsP6 synthesis directly from Ins(1,4,5)P3 [59]. Recently, it has been shown that PLC-generated InsPs are rapidly recycled to inositol, while the enzyme Inositol tetrakisphosphate 1-kinase 1 (ITPK1) phosphorylates Ins(3)P originating from glucose-6-phosphate, and Ins(1)P generated from sphingolipids, to enable the synthesis of InsP6 [60]. This finding suggests that InsPs with a higher number of phosphate units than InsP3 primarily come from endogenous myo-Ins, while lower InsPs, InsP3, InsP2, and InsP1, are provided from phosphoinositides hydrolysis. Currently, as many as 63 possible inositol phosphate esters have been identified as participating in biological functions [61]. However, inositol hexakisphosphate (InsP6) is the most represented among those phosphate derivatives [62], and it represents the starting brick to which phosphate groups are added, yielding inositol pyrophosphates (PP-IPs), in which one or two energetic di-phosphates bonds are aggregated around the six-carbon inositol ring [63]. Inositol pyrophosphates—and related kinases [64]—participate in the modulation of numerous biological functions in mammals, including morphogenesis, metabolic, and proliferation processes [65,66]. Remarkably, some studies indicate that inositol pyrophosphates have a regulator role in glucose and phosphate metabolism by finely tuning the balance between glycolysis and mitochondrial oxidative phosphorylation in ATP production [67,68] (Figure 1). 

Aside from the functions exerted by inositol-derived metabolites, myo-Ins, as such, has been described as a modifier of a number of processes involving endocrine signaling, morphogenesis, and reproduction, just to mention a few [69,70,71]. This evidence comes primarily from observational clinical studies, while experimental investigations—in vitro as well as in vivo—are still scarce. It is, therefore, hard to explain results obtained by adding myo-Ins in cell cultures or to the animal diet by using models that do not take into account the metabolic transformations to which the inositol molecule is committed after being absorbed by living cells. There is no doubt that we urgently need to explore the intracellular dynamic that governs myo-Ins transformation, and how the inositol network changes under different physiological and pathological circumstances. This is a very critical issue, as significant changes occur in myo-Ins metabolism in specific physiological and pathological conditions [72,73]. Furthermore, these alterations supposedly play causative roles in the pathogenetic process and are, therefore, plausible targets for medical interventions.

## 5. The Intriguing Role of D-Chiro-Ins

As previously recalled, under insulin stimulation, tissue-specific epimerases convert myo-Ins into its stereoisomer D-Chiro-ins. Unknown factors finely tune this irreversible reaction to deliver inositol isomers according to variable tissue-dependent needs [74]. The two isomers participate in the constitution of GPI anchors in which they represent the IPG core. IPGs incorporating either myo-Ins (IPG-A) or D-Chiro-Ins (IPG-P) are released upon stimulation of insulin by hydrolysis of GPI lipids located on the outer leaflet of the cell membrane. IPGs affect intracellular metabolic processes, namely by activating key enzymes controlling glucose metabolism [75]. It is worthy of note that IPGs both increase mRNA and protein expression of the insulin receptor substrate (IRS1), thus improving receptor-mediated insulin transduction. This effect requires the up-regulation of PI3K and the phosphorylated activation of AKT (pAKT) [76,77]. In addition, by increasing phosphate dehydrogenase (PDH) activity, IPGs enhance the oxidative metabolism of glucose, resulting in increased availability of ATP [78]. This is a pivotal tipping point, given that a decrease in the ATP/ADP ratio—together with insulin stimulation—amplify the enzymatic activity of IP6K1, thus leading to an increased synthesis of 5-diphosphoinositol pentakisphosphate (5-IP7). In turn, 5-IP7 reduces insulin sensitivity by preventing the interaction between AKT and PI3K, thus closing the regulatory loop. Moreover, the overactivation of IP6K1 is strongly associated with both insulin resistance and weight gain [79]. Therefore, we can hypothesize that the reduced availability of inositol-phosphoglycans could flatten ATP levels, thus enhancing IP6K1 activity and pyrophosphate synthesis. As expected, IP6K1 inhibition efficiently counteracts insulin resistance and obesity [80]. It is worth noting that myo-Ins dramatically downregulates IP6K1 activity by increasing ATP availability and miRNA 125a-5p release [81]. Therefore, myo-Ins-dependent modulation of IP6K1 can contribute to mitigating insulin resistance. 

The insulin-sensitizing capabilities of both myo-Ins and D-Chiro-Ins have been confirmed by several clinical studies [82,83]. However, it is unlikely that the improvement in insulin transduction could explain, by itself, the rescue of the ovarian and reproductive function obtained in PCOS patients treated with myo-Ins associated or not with D-Chiro-Ins. This statement helps explain why treatment with antidiabetic drugs, while improving several metabolic markers associated with PCOS [84], including androgen production, is still scarcely effective in ameliorating ovary function, as testified by the observed decrease in follicles number and quality after treatment with metformin [85].

Indeed, as previously recalled, D-Chiro-Ins given in isolation enhances androgen synthesis. Yet, as important as it could be, this effect does not explain in full the androgenic phenotype of PCOS patients, given that increased androgen production would, in turn, increase aromatase activity, leading to a proportional increase in estrogen release [86]. Theca cell androgens act not only as a substrate of estrogen synthesis, but also modulate FSH action via the activation of androgen receptors [87] and cAMP [88]. Furthermore, insulin and Insulin Growth Factor-1 (IGF-1) synergize with testosterone in amplifying FSH-enhancing effects upon aromatase expression [89]. Surprisingly, increased androgen release within the PCOS ovary is associated with reduced aromatase levels and impaired FSH transduction. Thus, D-Chiro-Ins not only enhances androgen synthesis but also impairs the estrogenic response of granulosa cells to FSH and androgens. This is a critical point, as it suggests that PCOS pathogenesis should integrate both an increase in androgens as well as a reduction in estrogens.

Intriguingly, clinical investigations in which myo-Ins is associated with very low doses of D-Chiro-Ins provided unexpected results, indicating that, while retaining the beneficial, insulin-sensitizing effects exerted by D-Chiro-Ins, the addition of myo-Ins allowed rescuing almost completely the ovarian function [90,91]. These apparently puzzling results suggest that myo-Ins and D-Chiro-Ins exert opposite effects on steroidogenic pathways within the ovary [92].

Treatment with myo-Ins (2 g twice a day) significantly decreases the LH/FSH ratio in the plasma of PCOS women [93,94], while supplementation with myo-Ins during in vitro fertilization allows the reduction in the doses of recombinant FSH administered [95]. Moreover, studies currently ongoing show that myo-Ins increases FSH receptor and aromatase synthesis in granulosa cells, probably through an FSH-independent mechanism of action, as myo-Ins raises the transcription of aromatase also in breast cancer cells without any previous FSH administration [96].

These data indicate that myo-Ins and D-Chiro-Ins play an opposite role in ovarian steroidogenesis, namely acting upon aromatase, androgen release, and FSHr expression. Again, let us outline that the downregulation of FSH receptors and aromatase in granulosa cells represents a hallmark of PCOS [97]. In this condition, it would be reasonable to expect that D-Chiro-Ins supplementation could worsen the clinical evolution of the syndrome. Such a hypothesis has been vindicated by a recent study performed in PCOS rodents treated with myo-Ins and D-Chiro-Ins according to very different ratios [70]. The report shows that only treatments with high myo-Ins concentrations (and low D-Chiro-Ins content) have proven to be effective in treating PCOS mice. In that study, animals were treated with different myo-Ins/D-Chiro-Ins formulas (from 5:1 to 80:1), and only mice receiving myo-Ins/D-Chiro-Ins in a 40:1 molar ratio made a fast and almost full recovery from PCOS signs and symptoms. Interestingly, formulas with higher D-Chiro-Ins content were demonstrated to be ineffective or even detrimental, ultimately worsening several PCOS characteristics. Noticeably, histological investigations revealed that the theca layer resulted in having almost double the thickness in controls and mice treated with high D-Chiro-Ins concentrations, indicating that this treatment induced functional androgenic hyperplasia. Conversely, treatment with myo-Ins/D-Chiro-Ins according to the 40:1 ratio (i.e., the physiological levels found in the blood) almost completely restores the normal architecture, with a physiological relationship between theca and granulosa cell layers.

Some observational studies have confirmed these findings, indicating that treatment based upon a proper myo-Ins/D-Chiro-Ins ratio (40:1) can efficiently counteract the foremost PCOS signs and symptoms [98]. Conclusively, the negative effect induced by D-Chiro-Ins upon ovary steroidogenesis ultimately vindicates the “paradox” argued by Unfer [99]. As previously recalled, myo to D-Chiro conversion is fostered by insulin, and in insulin-resistant patients, this would lead to a significant deficit of D-Chiro-Ins in many tissues. However, insulin resistance is not associated with impairment in the transduction of the insulin signal at the ovarian level, given that hyperinsulinemia still stimulates ovarian androgen production in PCOS, and can likely act similarly upon D-Chiro-Ins synthesis, thus finally impairing myo-Ins availability [15].

Remarkably, under insulin stimulation, the highest conversion rate of myo-Ins into D-Chiro-Ins is close to 9%, and was detected in the liver and muscle, two important insulin-sensitive areas, whereas in the heart and brain, this is less than 2%. These differences highlight that tissues display differential needs of both isomers to fulfill their specific metabolic demands. Insulin resistance can dramatically impair D-Chiro-Ins levels in many tissues, resulting in low intracellular levels of D-Chiro-Ins [20]. However, D-Chiro-Ins is unexpectedly increased in the follicular fluid recovered from PCOS ovaries, thus further indicating that PCOS ovaries display an astonishing insulin sensitivity and promote a sustained conversion of myo-Ins into D-Chiro-Ins [24]. To sum up, ovary tissue in PCOS subjects shows a significant deficit in myo-Ins content. Hence, this shortfall should be corrected to normalize ovarian functions.

Insulin resistance cannot explain the complexity of the polycystic ovary syndrome, given that this hypothesis is biased by assuming as a premise that PCOS pathogenesis essentially relies upon defective insulin transduction due to impaired availability of inositolphosphoglycans. This model leaves aside the opposite effects directly triggered by myo-Ins and D-Chiro-Ins upon ovarian steroidogenesis, and underestimates the specific insulin hypersensitivity displayed by the ovary in PCOS subjects. Obviously, the increased insulin secretion, as that observed in insulin-resistant subjects, can amplify the steroidogenic response of the ovary by raising D-Chiro-Ins levels in the gonads. Nevertheless, it cannot explain the abnormal insulin sensitivity displayed by the ovaries of PCOS patients, which show an increased presence of androgens and their related enzymes when stimulated by LH and insulin with respect to normal controls [100]. Thus, instead of being triggered by insulin resistance, PCOS should alternatively be considered as a syndrome of ovarian “hypersensitivity” to insulin [101].

Conclusively, insulin resistance is not a general feature of PCOS, since the prevalence of insulin alterations involves barely around 60% of patients [102]. Furthermore, it is puzzling why young PCOS women without insulin resistance still respond to myo-Ins treatment. In these women, by assuming that myo-Ins supplementation could “normalize” the myo-Ins/D-Chiro-Ins ratio within the ovary, we may confidently hypothesize that the increased bioavailability of myo-Ins—at levels overcoming the saturation threshold of ovarian epimerase—“rectifies” the inositol ratio, reactivating aromatase activity, restoring FSHr–based transduction, and ultimately reducing androgen synthesis [103,104]. Therefore, we suggest that the primary pathogenetic defects in PCOS could be an intrinsic abnormal insulin responsiveness in the ovary, leading to increased androgen synthesis altogether with reduced estrogen availability; meanwhile, other systemic factors (insulin resistance, deregulation of the hypothalamic-pituitary-axis, and obesity, just to mention a few) can potentiate this mechanism and exacerbate the overall clinical picture. The altered balance between myo-Ins and D-Chiro-Ins, resulting in a relative excess of the latter component, may represent a causative, relevant factor dramatically fostered by insulin secretion, even in the absence of an overt condition of insulin resistance.

## 6. Conclusions

The conclusive pathogenic hypothesis based on the preliminary hints provided by Larner [15] and Unfer’s [24] papers—vindicated by successive clinical evidence—suggests that D-Chiro-Ins alone stimulates androgen synthesis and decreases the ovarian estrogen pathway. On the contrary, myo-Ins activates FSH response and aromatase activity, finally mitigating ovarian hyperandrogenism (Figure 2). This would imply that inositols can be highly effective for those PCOS phenotypes featured by hyperandrogenism; otherwise, as in the case of phenotype D, they would not have a cogent therapeutic rationale.

When the two isomers are given in association—according to the physiological ratio of 40:1—patients could benefit from the insulin-sensitizing effects displayed mostly by D-Chiro-Ins. Yet, Larner’s study indicates clearly that we need not postulate insulin resistance to explain PCOS pathogenesis, given that insulin hypersensitivity is likely a common feature of PCOS ovaries. Indeed, even in the presence of a physiological insulin stimulus, the PCOS ovary synthesizes D-Chiro-Ins four times more than that measured in control theca cells. The increased D-Chiro-Ins within the ovary is detrimental to ensuring proper steroidogenic control, and this failure can easily explain why treatment strategies based upon high D-Chiro-Ins have been poorly effective. Within this perspective, two factors emerge as major determinants in PCOS: hyperandrogenism and reduced aromatase expression. Therefore, PCOS could no longer be considered a disease due to only increased androgen synthesis, without considering the contemporary downregulation of aromatase and FSH receptors.

However, that model raises several unanswered issues, which should be addressed by future studies.

First, why do different tissues show different myo-Ins/D-Chiro-Ins ratios, and how could differences in epimerase activity explain these data? Is there any difference in the structure of the D-Chiro-Ins epimerase among different tissues? In other words, what factors actually shape the different epimerase sensitivity to insulin in different histological contexts? Second, are there any other factors that prevent theca androgens from undergoing the physiological conversion into estrogens when ovaries are treated with D-Chiro-Ins? Third, how do inositols modify other endocrine receptors, LHr, AR, and ER, when both theca and granulosa cells are treated with both the two inositol isomers? Some preliminary reports show indeed that alternative splicing dramatically alters AR recruitment and androgen-induced expression of genes related to folliculogenesis in human granulosa cells, namely by impairing aromatase expression [105]. It is time to investigate how inositols could modify this picture.

Moreover, how does myo-Ins supplementation modify the complex inositol-related network, and how could inositol-related metabolites influence the overall picture? To obtain a reliable picture, we should be able to capture the whole metabolic fate of myo-Ins when added to a specific tissue/cellular context and correlate changes in inositol metabolism with the associated biochemical pathways [47].

Finally, maybe we should start thinking of tailored therapies for the different PCOS phenotypes, particularly for the non-hyperandrogenic D phenotype, which could be properly managed by natural, active compounds other than inositols.

Overall, these tasks are challenging. Integrated, multidisciplinary approaches—performed in in vitro and in animal studies—are warranted to address those issues, hence providing a compelling confirmation of the proposed pathogenetic hypothesis.

## Figures and Tables

**Figure 1 ijms-24-06296-f001:**
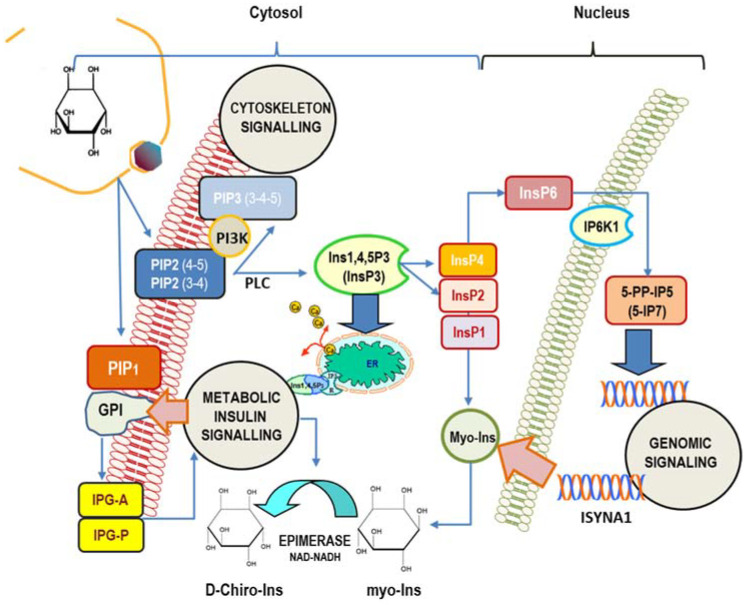
Main intracellular signaling pathways in which myo-inositol participates. Inositol is mostly involved in (1) cytoskeleton remodeling; (2) insulin transduction (noticeably through inositol-phosphoglycan containing myo-Ins (IPG-A) and D-Chiro-Ins (IPG-P) released from GPI (glycosylphosphatidylinositol-anchored proteins) and glucose metabolism); (3) genomic signaling, especially through the participation of inositol hexakisphosphate (InsP6) and pyrophosphates. In addition, myo-Ins—and its isomer D-Chiro-Ins—exert relevant effects upon the steroidogenic pathway in ovary cells. Cellular myo-Ins is supplied by external uptake and endogenously synthesized by ISYNA1 gene. PIP2 (3-4), 1-Phosphatidyl-1D-myo-inositol 3,4-bisphosphate; PIP2(4-5), 1-Phosphatidyl-D-myo-inositol 4,5-bisphosphate. PIP1, 1-Phosphatidyl-D-myo-inositol-phosphate; InsP1, inositol monophosphate; InsP2, inositol bisphosphate; InsP3, myo-inositol 1,4,5-trisphosphate; InsP4, myo-inositol-1,3,4,5-tetraphosphate; InsP6, myo-Inositol 1,2,3,4,5,6-hexakisphosphate; ER, endoplasmic reticulum; 5-PP-IP5, inositol 5-diphospho-1,2,3,4,6-pentakisphosphate (5-IP7); IP6K1, inositol hexakisphosphate kinase 1; PLC, phospholipase C.

**Figure 2 ijms-24-06296-f002:**
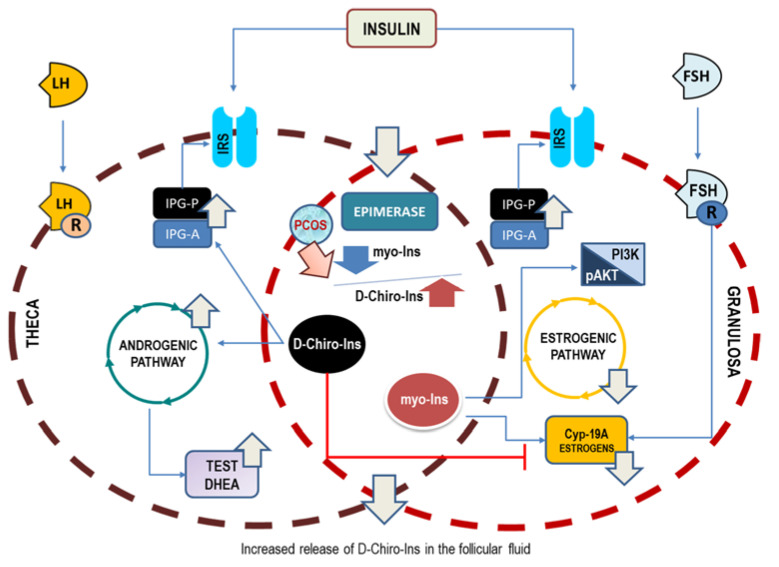
Opposite effects upon ovary steroidogenesis of myo-Ins and D-Chiro-Ins. Insulin stimulation promotes the release of phosphoglycans, containing either myo-Ins (IPG-A) or D-Chiro-Ins (IPG-P). Moreover, insulin activates the epimerase-dependent transformation of myo-Ins into D-Chiro-Ins. Test, testosterone; DHEA, dehydroepiandrosterone; Cyp-19A, aromatase.

## Data Availability

The data presented in this study are available in the article.

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
