# Peer review of "The Role of Inositols in the Hyperandrogenic Phenotypes of PCOS: A Re-Reading of Larner’s Results"

_ijms, 2023, doi:10.3390/ijms24076296_

Round 1

Reviewer 1 Report

This is an interesting review article that focuses on a topic not well-known within the PCOS field: the possible role of inositol variants in driving androgen hypersecretion in response to insulin. In my opinion it represents a worthwhile addition to the PCOS literature.

Comments/critiques

For some reason the in-text references are in Roman numerals (except for line 174) while the bibliography at the end are numbered with the usual Arabic numerals. This made reviewing a bit difficult and needs to be sorted.

I can’t see any reason for the use of capitals in the terms myo-inositol and d-chiro-inositol. They are not proper nouns or brand names.

Line 43: is PCOS really the most common endocrinological disorder of woman? What about dysmenorrhea? Maybe “the most common cause of anovulation” would be safer.

Figure 1 seems very ‘busy’ to me. About half of the images contain text and details too small to read; simpler images would be better. I’m at a loss to know what the small uterus with an arrow pointing to the big uterus is supposed to indicate. Polycystic ovarian syndrome (in the figure legend) should not be capitalized.

Line 84: correct ”inhibition insulin”

Line 86: correct “PCOS ovary”

There are not very many preclinical studies cited (although I found reference #40 interesting) (line 170). I wonder if this is because most animal PCOS models that are characterized by even mild insulin resistance involve chronic treatment with testosterone or DHT (by contrast, mouse PCOS models programmed by brief prenatal DHT or AMH exposure don’t show a metabolic phenotype). Therefore, it becomes impossible to test interventions to correct insulin-induced hyperandrogenism in these models, since the hyperandrogenism is exogenous. Maybe this is getting off the track, but might possibly be worth a mention.

Figure 3 is perhaps the most relevant figure to the review topic, but I think the key point is missed: the effect of PCOS on myo-inositol and d-chiro-inositol ratios. Could this be built into the figure somehow? Or does it require another figure? It takes quite a bit of reading to understand this key point from the text, so it would be good to use the figures to help the reader appreciate this at a glance.

Reviewer 2 Report

It is a well written review article

Suggestion to include some basics causes and mechanism of PCOS

Mention other causes of PCOS (lifestyle causes and its link to hormones)

Reviewer 3 Report

The publication Fedeli et al. is very interesting in terms of the correct interpretation of the effects of inositols. Despite the use of results published in a relatively distant time in the light of today's knowledge, the results obtained by the Larner’s team take on a new meaning. This review is not just another presentation of already known facts, but in my opinion a successful attempt to interpret and discuss the results, which thus gain in value.

However, I have some notes that authors should change before publishing the article:

·     Most importantly, there are differences between the marking of citations in the text and the same items in the bibliography. Therefore, I could not check the correctness of citations, which I consider as a serious disadvantage. I can only guess the connection between Roman and Arabic numbers in this publication. Regardless of whether it was intentional or a mistake, authors must correct citations in such way that the citations will be identical with the literature. Citations in the text should be clearly and uniformly marked. The reader should be able to easily find and check the correctness of these citations, which will enable a full understanding of the publication.

The bibliographical list is rather unified, but there are some small mistakes, so I would suggest checking the whole thing (e.g., items 1 and 2 are different from the rest - note the commas in names and dates in bold).

·        ·     ChapterInositols in mammals” is too long and too general. I would suggest a division into 2 or 3 subchapters that would take into account the role of insulin or the impact of individual inositols on LH/FSH ratio and FSH receptor.

    ·     There seems to be something missing in the illustration of the reproductive organs in Fig. 1, e.g. arrow or signature? Besides, what do dashed arrows and continuous arrows mean? It should be described in the text below the Fig.1.

     ·     There are some minor mistakes in the text, e.g. tipos: line 174, 310; double space: line 22, 72, 247, 252 or no space: line 74. Then, I would suggest checking the whole text.
